# The Integral Formation of Catholic School Teachers

Amy E. Roberts [1,*] and Gerard O'Shea [2]

1 Department of Theology/Catechetics, Franciscan University of Steubenville, Steubenville, OH 43952, USA
2 School of Education, The University of Notre Dame Australia, Fremantle, WA 6160, Australia
* Correspondence: aroberts@franciscan.edu

**Abstract:** The Catholic Church has a long history of conducting schools as part of its mission to evangelize. This paper will contend that in order for teachers to implement the evangelistic mission of Catholic schools, they themselves need an integral formation that puts every dimension of their human nature—body, emotions, will, and intellect—in ongoing communion with Christ and His Church. A brief examination of the impact of secularization in the United States on the Catholic school mission indicates that teachers are inadequately formed to fulfill that mission. Contemplative practice, a common faith formation practice used for Catholic school teachers, will be evaluated as insufficient for achieving its goal because it does not fully account for the way God created human beings. Contemplative practice relies heavily on the work of John Dewey, who applied inadequate anthropological principles to the task of human learning and teacher education. By contrast, faith formation efforts that account for human nature engage both the *intellectus* and the *ratio*, and in so doing engage the teacher's whole integrated person. Teacher faith formation can facilitate the teacher's encounter with God, allowing Him to form her, by providing analogical encounters with Him through the transcendentals and sacramental encounters with Him in the liturgy.

**Keywords:** Catholic schools; faith formation; teachers; Christian anthropology; transcendentals

## 1. Introduction

The Catholic Church has a very long history of conducting schools as part of its mission in the world flowing from what is known as the *Great Commission*: "Go therefore and make disciples of all nations, baptizing them in the name of the Father and of the Son and of the Holy Spirit, teaching them to observe all that I have commanded you . . . " (Matthew 28:19–20). In every period of the Church's history, this imperative of evangelization has remained the foundation of the Catholic school's mission, right up to the most recent teaching from the Congregation for Catholic Education in January 2022, *The Identity of the Catholic School for a Culture of Dialogue* (Congregation for Catholic Education 2022, #6). Furthermore, the 2020 document, *Directory for Catechesis* reiterated the consistent teaching of the Church regarding the Catholic formation of teachers: "It is required that the teachers be believers committed to personal growth in the faith, incorporated into a Christian community desirous of giving the reason for their faith through professional expertise as well" (Pontifical Council for Promoting the New Evangelization 2020, #318).

This paper will contend that in order for teachers to implement the evangelistic mission of Catholic schools, they themselves need an integral formation that puts every dimension of their human nature—body, emotions, will, and intellect—into ongoing communion with Christ and His Church. It will begin by examining contemplative practice, commonly used in the faith formation of Catholic school teachers and educational leaders of the United States. It will proceed to argue that an adequate anthropology empowers teachers to fulfill the mission, because it presumes a particular vision of human learning, a vision that contemplative practice cannot adequately fulfill. It will then evaluate the vision of human learning articulated by the American educational philosopher John Dewey, whose ideas permeate teachers' professional formation in the United States. It will maintain that

the nature of the human person is such that learning must integrate all her[1] powers to discover truth. Thus, the paper will argue, human learning must integrate all the person's powers to arrive at the truth and to make a willing commitment to God. Finally, the paper will propose intentionally and strategically applying the transcendentals to teacher faith formation in order to form the teacher's whole person through analogical encounters with God.

## 2. Historical Context

Catholic schools in the United States inherit and implement the Church's mission and vision for its schools. This mission, reiterated in *The Identity of the Catholic School for a Culture of Dialogue*, extends Jesus' Great Commission by educating the whole person for union with God. Rather than simply responding to a civic or social need to educate young people, Catholic schools "participate in the evangelizing mission of the Church" (Congregation for Catholic Education 2022, #30) for the sake of "proposing" the Gospel to the next generation (Benedict XVI 2008). Furthermore, in educating the whole person, Catholic schools help students "come to the fullness of Christ's life" by interweaving evangelization with integral human development (Congregation for Catholic Education 2022, #13). Because the Church is both Mother and Teacher, its schools "must be concerned with the whole of man's life, even the secular part insofar as it has a bearing on his heavenly calling" (Congregation for Catholic Education 2022, #10). In fulfilling this mission, the Catholic school endeavors to form students in the Catholic faith. Faith, in the Catholic understanding, is a gift of God and a human response to that gift, so that the recipient of the gift in turn "gives personal adherence to God and freely assents to the whole truth that God has revealed" (Catechism of the Catholic Church 2009, Glossary).

However, growing secularization in the United States presents challenges to Catholic schools' fulfilling the mission. Polls (such as Smith 2019; Diamont 2020; Smith 2022) continually show low levels of faith practice and belief among Americans, including American Catholics, implying that fewer people want to enroll their children in Catholic schools for the sake of forming them in the faith. Enrollment in Catholic schools in US has declined since peaking in the 1960s at about 5.2 million students to fewer than 1.7 million students in the 2021–2022 academic year, indicating that fewer students have the opportunity to learn the Catholic faith in Catholic schools than did 60 years ago (National Catholic Education Association 2022a, 2022b). Catholic schools have fewer religious and clerical teachers to hand on the Catholic faith, in religion class and throughout the whole curriculum. Consequently, Catholic schools are relying heavily on lay teachers to fulfill the mission.

It is clear, however, that many lay teachers have been inadequately prepared to fulfill the Catholic school mission. In 1997, the United States Conference of Catholic Bishops released a report stating that after reviewing a number of commonly used religion texts, they had found ten doctrinal deficiencies that consistently permeated these texts. Corrected texts were not available until August 2011. Catholic teachers, formed in the faith as students in Catholic schools or parish schools of religion, often suffered inadequate faith formation as a result of these deficiencies.[2] Those who did not supplement their inadequate formation could not adequately hand on the faith to their own students. Furthermore, most Catholic school teachers are professionally prepared as educators in secular institutions that do not address spiritual formation. As a result, these teachers do not receive professional formation that equips them to implement the evangelizing mission of Catholic schools. It seems, then, that the current context of Catholic schools in the United States is such that teachers need faith formation in order to carry out the mission effectively. How then, can we begin to address the issue of Catholic identity among teachers who need to be formed adequately to fulfil their mission?

### 3. Contemplative Practice, Commonly Used in the Faith Formation of Catholic School Teachers

One common practice in teacher faith formation, contemplative practice, is insufficient for the purpose of establishing authentic Catholic identity in a school because it relies on subjective experience that may or may not be informed by objective truth.[3] Advocated for by Merylann Schuttloffel of Catholic University of America (Schuttloffel 2013), promoted by the National Catholic Education Association and in continued use by Catholic educators (Fussell 2021), contemplative practice is a "metacognitive approach to support [Catholic educators'] decision-making, communication, and interactions with members of the school community" (Schuttloffel 2016, p. 179). Through contemplative practice, Schuttloffel encourages teachers and school leaders to "think about their own thinking" and its effect on the Catholic identity of the school (Schuttloffel 2013, p. 84). She states that "explicitly gospel values, Catholic theology, and Church tradition" inform activities and decisions in the school, as teachers and leaders "consider personal values, beliefs, and philosophy [to] think about *why* they think the way they do" (Schuttloffel 2013, p. 83). She presumes that "the values, beliefs, and philosophical anchors are taken from gospel values, Church teaching, our Baptismal call for evangelization, and the Catholic Church's intellectual tradition" (Schuttloffel 2013, p. 83). These anchors embed teachers' "interpretive reflection . . . in a Catholic worldview [that is] rooted in a well-formed Catholic identity" (Schuttloffel 2013, p. 83). Schuttloffel's vision of contemplative practice acknowledges that "who" a teacher or leader is, affects the decisions they make and the example they set for students, an indispensable outcome for any teacher faith formation. However, Schuttloffel does not allow for the fact that the prior professional and faith formation that teachers bring to Catholic schools does not always include the influence of objective Church teaching and practice. She herself states that contemplative practice assumes that teachers have received or currently receive "explicit faith formation" especially in the sacraments and parish life (Schuttloffel 2013, p. 84). Since polls repeatedly show low levels of belief and practice of the Catholic Faith, Schuttloffel's assumption seems unfounded.

Moreover, contemplative practice does not account for the effects of the Fall on the teacher. Traditionally, the Catholic Church has defined the Fall as the original sin committed by the first human beings, causing all human beings to be conceived without God's grace, bringing about a "darkened intellect" and a "weakened will" that are not completely removed when God's grace is given in Baptism (Catechism of the Catholic Church 2009, Glossary and #1264, #1426, #2515). These effects imply that teachers do not necessarily know, understand, or believe the truths of Catholicism without those truths being taught to them, and they do not always act virtuously or embrace Catholic values. A brief examination of the world today suggests that all human persons, teachers included, suffer from these effects. While "thinking about one's thinking" gives teachers opportunity to evaluate the effectiveness of their values and practices, the teachers' evaluation is subject to ignorance, misinterpretation, or even moral failing.

Finally, contemplative practice presumes that teachers are responding to their call to union with God and asking for His grace. Polls showing American Catholics' poor and infrequent attendance at Sunday Mass means formators cannot assume teachers seek God or His grace and truth apart from what faith formation efforts provide. While it might be safe to assume that a poll of Catholics teaching in Catholic schools would show a higher percentage of regular Mass attendance relative to all other American Catholics, it seems probable that a significant number of Catholic teachers do not avail themselves of God's grace in an active sacramental life, much like the rest of the culture.

Given the current state of American Catholics' faith belief and practice (as highlighted in the previous section), Schuttloffel seems excessively optimistic to assume that teachers and school leaders have the foundational formation that they need to carry out contemplative practice fruitfully in their schools. Nevertheless, Schuttloffel's model is widely utilized in faculty faith formation. The implications of the current situation in Catholic schools means that faith formation for teachers will have to make up for the aforementioned

deficiencies. Contemplative practice, in and of itself, does not adequately provide what teachers need to fulfill the mission of Catholic schools. To adequately provide what teachers need, we must first examine the nature of the human person.

## 4. Adequate Anthropology Empowers Teachers to Fulfill the Mission

Designing faith formation efforts that account for human nature empowers teachers to fulfill the Catholic school mission.[4] Human nature is characterized by relationality, the spiritual (non-material and therefore non-sensate)[5] powers of intellect and will, and the union of the body and the soul. With regard to relationality, the human person is created in the image of the Trinity—three persons, one God. The nature of the Trinity as a communion of persons who exist in relation to one another and in communion with one another implies that teachers, created in God's image, are created precisely for communion with God and with other human beings. In fact, "the deepest core of the human person lies in [her] capacity for relationship" (Granados 2010, p. 295), a capacity that is fulfilled first by relationship with God and then with other human beings (Ratzinger 1990, p. 445 and Congregation for Catholic Education 2007, #8). Accommodating the communal aspect of human nature when forming teachers includes forming them for their final end, which is union with God (Second Vatican Council 1965, #1). This end informs the fundamental mission of Catholic schools—educating the whole person for union with God. The teacher will necessarily have to have received this kind of formation in order to form her students accordingly. Thus, teacher faith formation must include fostering her relationship with God as well as with her human community.

Communion between persons demands all the human powers: not just the emotions and the body, but also intellect and will. Inevitably, human beings must know the truth about those with whom they are in relationship in order to cultivate that relationship, and they must do good for them. Experience demonstrates that knowing very little about or doing harm to another person weakens the relationship. The human intellect seeks truth while the human will is directed toward the good (Aquinas 1922, I, 16, 1)—including the true and the good about the other person. Truth and goodness, as stable realities existing outside the person, originate in God, who <u>is</u> Truth (Aquinas 1922, I, 16, 5) and Goodness (Aquinas 1922, I, 6, 1–2). For teachers, then, the intellect and the will are spiritual powers that enable them to seek God Himself when seeking truth and goodness (Aquinas 1922, I, 79, 1 and I, 82, 3). Consequently, teacher faith formation needs to form the intellect and will to seek God Himself, as well as to seek the truth about God and to imitate Him in goodness. Furthermore, the union of body and soul, material and spiritual, implies that the spiritual powers of intellect and will require a body to be exercised. For human beings, bodily experiences inform the intellect, shape the will, and even stir the emotions—all of which are important for facilitating communion with God (Granados 2010, p. 297). The Incarnation demonstrates *par excellence* that the bodily and the concrete opens the door for communion with the spiritual and the divine (O'Shea 2013, p. 457). This indicates that teachers' faith formation needs to address their bodies, affectivity, will, and intellect as they are invited into deeper communion with God. While Schuttloffel's contemplative practice rightly accounts for bodily experience, it assumes rather than provides for the bodily experiences that give communion with God, namely, the sacraments.

The above has hinted at a crucial point for teacher faith formation: the human powers do not act independently but rather are integrated with one another. Aquinas argued that in the human person, the intellect moves the will *and* the will moves the intellect together with all the powers of the soul (Aquinas 1922, I, 82, 4). The will can move the sensitive power (Aquinas 1922, I, 81, 3), the ability of the bodily senses to perform their designated function and thereby to perceive accordingly what is presented to it (Aquinas 1922, I, 80, 1–2). Conversely, the sensitive appetite, the inclination to perform or not perform the functions of the senses, can also move the will (Aquinas 1922, I-II, 9, 2). These movements of the soul are expressed by the body (Aquinas 1922, I, 81, 3), even as bodily experience influences the soul. John Damascene referred to this continual movement between faculties of the

soul using the Greek word, *perichoresis* (a dance—John of Damascus 2022, sct. 57–58). His primary concern was to offer an explanation of the operation of the Holy Trinity, in which the Persons mutually indwell one another in an indivisible unity while simultaneously distinct from each other (Catechism of the Catholic Church 2009, #253–255). He further argued that, since human beings are created in the image of God who is a Trinity, there is an analogical similarity in the way they operate. In other words, the human person functions in a way that is analogous to the Trinity (Fourth Lateran Council, *Canon* 2).[6] This fruitful idea has found its way into much contemporary theology. Moreover, while it is true that the separation of human beings from God that resulted from the Fall damaged human capacities, the wounded powers are not isolated from one another.

This perichoretic integration of the human faculties implies that our learning, and thus teacher faith formation, must account for more than just the intellect. The notion *perichoresis* and its influence on contemporary theology has already been noted. Other ancient and medieval philosophers such as Aristotle and Aquinas, and the contemporary German authors, Joseph Pieper (Pieper [1948] 1998, pp. 9–20) and Joseph Ratzinger (later Benedict XVI; see for example Ratzinger 2002),[7] add further dimensions to this idea. Humans do not learn by exercising their intellect alone, but through the integration of all their powers. In this view, the human intellect includes the *ratio* (or reason) and the *intellectus*. The *ratio* is the ability to discover truth through abstraction, analysis, logical reasoning, and the like. The *intellectus* enables the intellect to see the truth by "simply looking" (*simplex intuitus*), by taking in input from the bodily senses, from the sensitive appetite, from the whole of a person's experience. The *intellectus* informs the *ratio* of what it has observed. The *intellectus* receives truth, and the *ratio* uses the truth received from the *intellectus* to do the difficult work of reasoning its way to rational knowledge. The knowledge received by the *intellectus* is "connatural knowledge," an understanding acquired through the integration of all the human powers and a participation in some way with the nature of the object that is known (Taylor 1998, p. 64). The *ratio* cannot produce connatural knowledge, but rather connatural knowledge forms the basis for some significant activity of the *ratio.*

Distinguishing and clarifying the roles of the *intellectus* and the *ratio* is critical for the faith formation of teachers. In his 1979 document *Catechesi Tradendae*, Pope John Paul II stated that the aims of catechesis are "understanding" and conversion. The word he chose for "understanding" was *intellectum*, not *ratio.* The goal of understanding in faith formation is not limited to the knowledge attained through analysis, discursive reasoning, and active labor. Rather, faith formation has the goal of connatural knowledge, an integrated knowledge of the object known, and even a "participation" in the object known—in this case, God. It might be said that the *ratio* can lead to teachers' knowledge <u>about</u> God, but the *intellectus* can lead to teachers' knowledge of God Himself.

Mother Veronica Namoyo Le Goulard, a Poor Clare nun, recounts a personal experience that illustrates the *intellectus* at work. Le Goulard was born to atheist parents who decided to raise her in Morocco so that "nobody could speak to me of God, and no one could influence the development of my mind with oppressive superstition" (Le Goulard 1993, p. 23). But when she was three years old, she witnessed a glorious sunset, whose beauty overwhelmed her. She recalls:

> "I was caught in limitless beauty and radiant, singing splendor. And at the same time, with a cry of wonder in my heart, I *knew* that all this beauty was created, I knew God. This was the word that my parents had hidden from me. I had nothing to name him: God, Dieu, Allah or Yahweh, as he is named by human lips, but my heart knew that *all was from him* and him alone and that he was such that I could address him and enter into relationship with him through prayer . . . Not once could I dismiss this experience, whatever my intellectual doubts might have been in the following years". (Le Goulard 1993, p. 30)

Le Goulard's experience of observing the sunset, taken in through simple intuition, prompted the understanding of the *intellectus* to recognize the Creator of this beauty, even though she had never been told of such in her conscious memory. In spite of her *ratio*

raising objections in later years, she could not forget or dismiss the understanding she experienced as a small child. This understanding of the *intellectus* gradually resolved those objections, and ultimately connected her *ratio* to her concrete experience.

Countless faith formation efforts account only for the *ratio*. In contemplative practice, teachers "think about their thinking," about their values, about why they think as they do. Another frequent practice in faith formation focuses on strictly instructional approaches to teaching doctrine. Arguably, the hyper-emphasis on the *ratio* in these faith formation practices has led to many teachers knowing about God and knowing about themselves, but not knowing God. Jose Granados argues that "Christianity is not a cold reflection on a purely spiritual idea, but is rather a living experience of God that embraces the whole of man's being," which the *ratio* alone cannot accomplish. For contemplative practice to effectively form teachers in such a way as to enhance the fulfillment of the mission of Catholic schools, teachers must have experiences that present God Himself to their whole person.

A vision of human learning, and therefore a vision for teachers' faith formation, follows from an understanding of human nature. It is necessary, then, to examine and evaluate the anthropology that undergirds the vision of human learning that permeates most American teacher training today.

## 5. The Vision of Human Learning Articulated by the American Educational Philosopher John Dewey

Predominant understandings of human learning often do not account fully for this integrated, Christian vision of the human person. The views of American educational philosopher John Dewey can be counted among these. Dewey significantly influenced the professional formation of teachers in the United States through his teaching and writing in the late nineteenth and early twentieth centuries. Today, educational philosophy textbooks in most American universities quote heavily from Dewey. As we shall see, Dewey's principles arguably influenced Schuttloffel's contemplative practice. Like all philosophers, Dewey derived his philosophical principles from philosophers who preceded him. Because of his pervasive influence on teaching and learning United States classrooms, which is often imitated in faith formation efforts, this section will analyze and evaluate some aspects of Dewey's vision of human learning and the anthropological presumptions that undergird them.[8]

Dewey used and valued an empirical method of learning (see for example Dewey 1965c, pp. 158–59, 166–68), which only measures the material but cannot measure the spiritual. In this respect he can trace his philosophical roots to Rene Descartes, whose attempt to scientifically "disprove" religious skepticism prompted him to try to "demonstrate truth in an exclusively rationalistic and systematic way" (Taylor 1998, p. 108). Ultimately, Dewey's Cartesian belief led him to dismiss the spiritual dimension of the human person since it could not be demonstrated empirically. The result of Dewey inserting Descartes' position into mainstream American education is twofold: the common challenge to "prove" spiritual matters by empirical method, or the reactionary response of accepting spiritual assertions on "blind faith" divorced from reason. Contemplative practice reflects this divorce by failing to explicitly form the teachers' spiritual dimension, prompting teachers to "think about their own thinking" rather than cultivating their relationship with God. Furthermore, Schuttloffel's contemplative practice does not account for the spiritual effects of the Fall, which cannot be scientifically "proven" by any empirical methodology.

Another hallmark of Dewey's educational philosophy is his view (like that of Kant) that things cannot be known in themselves, that their nature is assigned to them by the observer rather than inherent in their being (Dewey 1965b, pp. 94–95). His view developed from applying to learning Darwin's theory of evolution (Dewey 1965a, pp. 1–2). Dewey held that knowing evolves according to the influence of the learner's environment. For Dewey, truth changes and so human beings should not bother searching for objective truth (Dewey 1965c, p. 164). Contemplative practice relies on the subjective experience of the

teacher, prompting the teacher to assign meaning to their experiences and to discover her values. To the extent that her experiences and values have been formed by the objective truth conveyed by the Church, the teacher may be able to benefit from contemplative practice in a way that helps her to foster her communion with God. However, if the teacher is unaware of objective truth or has explicitly rejected it, contemplative practice is unlikely to facilitate her own communion with God or that of her students, hindering her ability to fulfill the Catholic school mission.

Dewey further held that experience and empirical method determine truth rather than discover it. If there is no objective truth, then no search for it will lead to its discovery. Instead, human beings observe, recall, inquire, define, research, judge, test and the like in order to "reconstruct" thoughts and to solve problems. In Dewey's view, thinking is active, doing, and "intelligence is a method" (Dewey 1929, p. 220). He denies an "*a priori* form of non-reflective knowledge, one which is immediately given" (Dewey 1929, p. 221). He believed students needed to be taught *how* to think, rather than *what* to think (Dewey 1925, pp. 177–78). Dewey's position finds its roots in Kant's belief that human knowing requires "active mental effort . . . [knowing] is *activity,* and nothing but activity." (Pieper [1948] 1998, p. 10). For Kant, all human knowing is work, and knowing is only valuable to the extent that it has been acquired through the kind of work that Dewey advocated for student learning.[9] This anthropological position has informed contemplative practice, which appears to rely on the teacher's effort to distill from personal experiences all that is needed for her to cultivate her relationship with God and thereby contribute to the Catholic school mission.[10]

Ultimately, Dewey's anthropology, which undergirded his educational philosophy, eliminated the spiritual dimension of the learner, negated the possibility of finding objective truth, and held that all learning must be the result of effort, the *ratio.* While the above is a simplified and incomplete summary of Dewey's principles, he nevertheless developed his theory of learning out of anthropological principles that do not account for all that a human person is. The nature of the human person is such that learning must integrate all her powers to discover objective truth—a stable reality that includes the material and the spiritual. No less does anthropology factor into teachers' faith formation. All of her powers need to be integrally formed for the purpose of fulfilling her need for union and communion with God.

### 6. An Integrated Approach to Arrive at the Truth and to Make a Willing Commitment to God

Teachers' faith formation will be more complete and more likely to foster union with God if it integrates the *intellectus* as well as the *ratio.* In contrast to Kant, Dewey, and others, ancient and medieval philosophers held that the *ratio* can perform its work of knowing when it also participates in the *intellectus*, which understands by receiving from the senses and from the sensitive appetite, from the will, and sometimes by the direct inspiration of the Holy Spirit.

The *intellectus* can give the *ratio* a participatory and experiential kind of knowing by taking in data from the bodily senses as well as by means of direct inspiration. The senses perceive the concrete realities of the teacher's experience and the objective truth of those realities. Jose Granados argues that for embodied human beings, experience requires bodily participation in the world, a requirement indicative of how teachers can encounter God: "The Christian tradition connects the bodily senses to the experience of the divine . . . this connection means that the bodily senses are open to transcendence, and that the spiritual senses grow out of the bodily ones" (Granados 2010, pp. 293, 298). This position accounts for the union of the body and soul, the material and the spiritual, in human beings and roots teachers' *ratio* in bodily experience of reality. Connection between bodily and spiritual senses means that teacher can encounter God through the sacraments, allowing God to vivify all her human powers with His grace. For this reason, teacher faith formation must include provision of the sacraments to facilitate their union with Him.

Unlike focusing on the *ratio* alone, integrating the *intellectus* allows for and evokes teachers' capacity for connatural knowledge. The bodily senses stimulate the sensitive appetite, more commonly known as the emotions. Teachers, like all human beings, love and desire that which they perceive as attractive and beautiful; they are repulsed and fearful of that which they perceive as harmful or ugly. When God and the things of God are presented to the teachers attractively, they are drawn to a desire for union with Him and are drawn by their response of love to Him. But targeting teachers' faith formation to the *ratio* alone demands significant intellectual labor from the teacher that Pieper contends results in a weariness and hardness of heart that stifles the teacher's desire for God (Pieper [1948] 1998, p. 14). To illustrate, the English philosopher John Stuart Mill describes his experience of being educated strictly according to the *ratio*:

> "The habit of analysis has a tendency to wear away the feelings: as indeed it has, when no other mental habit is cultivated, and the analyzing spirit remains without its natural complements and correctives . . . I was thus left stranded at the commencement of my voyage with a well-equipped ship and a rudder, but no sail; without any real desire for the ends which I had been so carefully fitted out to work for: no delight in virtue, or the general good, but also just as little in anything else . . . There seemed no power in nature sufficient to begin the formation of my character anew, and create in a mind now irretrievable analytic, fresh associations of pleasure with any of the objects of desire". (quoted in Taylor 1998, pp. 113–14)

Pieper and Taylor, relying on the anthropology of Aquinas, hold that the *intellectus* receives, whereas the *ratio* labors. Laboring without receiving wearies the human heart, rendering it "stony" and unable to receive (Pieper [1948] 1998, p. 14). By contrast, when teachers experience satisfaction, joy, delight in concrete objects that stimulate a desire for more, the Holy Spirit can act (Granados 2010, pp. 296, 298). When the teacher's heart is not hardened by her own insufficient efforts to learn, it is free to allow God to stir her affectivity, her love for Him and His revelation. Love for God prepares the teacher to receive and embrace His revelation.

Love for God, stirred by integrating the *intellectus,* facilitates teachers' freely willed obedience to God. In creating human beings with free will, God necessitated that He would not violate that freedom by forcing a response to Him. However, Aquinas viewed the role of the sensitive appetite is to move the will to act. Thus, teacher's love for God moves her to respond to Him by choosing to follow Him and to imitate Him, as His commands make known. The teacher's *ratio* demonstrates the goodness of God and His commands, but creating the conditions for allowing the *intellectus* to receive allows the teacher to freely and willingly follow God.

None of the above should suggest that the *ratio* is unimportant and to be dismissed in teacher faith formation. Rather, it should show the danger of targeting the *ratio* alone, for such an exclusive focus diminishes the formation of the teacher's spiritual dimension and her communion with God. To integrate the *intellectus* in teacher faith formation, formators can intentionally and strategically apply the transcendentals.

## 7. A Way Forward: Applying the Transcendentals to Teacher Faith Formation

The Italian educator Maria Montessori, and those who have applied her ideas in the area of faith formation (particularly Sofia Cavalletti and Gianna Gobbi) made a strong case for the view that it was not only intellectual, discursive, input that mattered in the formation of the human person (see Cavalletti 1992; Gobbi 2000). Montessori used explicit Catholic language to explain what was happening in the development of a human being. She referred to it as "progressive incarnation in which the spirit and flesh are brought into an ever more perfect harmony" (Standing 1957, p. 210). Standing noted that Montessori's observations show an "unexpected affinity" with Aristotelian philosophy (Standing 1957, p. 212). This could be better described as an affinity with Thomist philosophy. It is, indeed, here that we must seek Montessori's perception of what Thomas Aquinas referred



to as *simple intuition* (Aquinas 1922, I, 59, 1). That is, in the field of religious formation, human beings need more than reflection and contemplative practice such as Schuttloffel recommends. They need an opportunity to receive inspiration directly from God.

Montessori emphasized the use of the senses, famously insisting that practical life activities gave students the opportunity for reflection and more importantly, spiritual contemplation (Standing 1957, pp. 212–28). She demonstrated that these contemplative opportunities worked better if they were associated with unhurried movements of the body (dusting, polishing, flower arranging and the like). In this way, she gave us a key to unlock access to the *intellectus*.

It seems that adults also respond to these opportunities for contemplation. The starting point is an acknowledgment of the unity of body and soul activating the perichoretic operation of every human faculty. It is never just about engaging the mind in abstract thought. Through the Church, teachers encounter God in the sacraments, where He appears behind the veil of signs perceptible to the bodily senses. However, faith formation for teachers can also provide analogical encounters with God through the transcendental properties of being present in every existing thing according to its own nature—beauty, goodness, and truth (Willey et al. 2008, p. 65). These transcendentals are manifested in concrete reality and perceived through bodily senses. They originate in God, the Source of all being, and while material manifestations of the transcendentals are not God, they do point to God, allowing teachers to encounter Him in the world He created. Balthasar, echoing Aquinas, links each transcendental with a Person of the Trinity, so that beauty points the teacher to the Son, goodness leads to the Holy Spirit, and truth directs to the Father (see McInerny 2020; Aquinas 1922, I, 39, 8). Such linking does not imply that one Person of the Trinity monopolizes the transcendental linked to Him. Rather, because the transcendentals are united within themselves (much as the Persons of the Trinity are united and indivisible), this linking makes distinctions that can assist in facilitating teachers' communion with God. God engages teachers' spiritual and relational dimension through these analogical encounters with Him in the transcendentals, and He thereby forms them for their final end of union with Him.

God also acts upon teachers' human faculties of intellect and will through their engagement with Him in the experience of the transcendentals. Beauty permeates bodily senses, moving the heart to love the good, which opens the mind and makes it receptive to the truth. This order is consistent with Aquinas' view of the integration of the human powers and the procession of human learning. It also illustrates the transcendentals' contribution to the *intellectus*. By "simply looking" at material manifestations of beauty, goodness, and truth, the *intellectus* can participate in the reality of it, and with God as the Source of it.

Furthermore, the theological virtues operate upon teachers' faculties, taking them beyond their natural capabilities alone. The theological virtues are gifts from God, but each can be strengthened by a corresponding transcendental:

> "Our yearning for truth leads us to seek the kind of truth that is beyond our intellectual capabilities—*faith*. Beauty is perfected by the incomparable harmony and perfection to be found in God, which is the wellspring of *hope*. Human goodness is called by Christ to the higher goal of *supernatural charity*". (O'Shea 2018, p. 235)

Since the mission of the Catholic school is more than just instruction in teachings of the Church, but educating the whole person for union with God, teacher formation should address the virtues, and can do so through the integration of the transcendentals.

Strategically applying the transcendentals to teachers' faith formation allows teachers to analogically encounter God in ways that transform her whole integrated person to become "like God." Faith formation efforts, then, incorporate beauty to attract teachers through their bodily senses. Examples of beauty can include God's creation and man-made images of it, as well as sacred and religious art, music, literature, and the dramatic arts. Each of these forms of beauty—to the degree that they are also good and true—

communicate religious truth in ways that move teachers to embrace the good and flee evil. By participating in beauty, through looking, listening, reading, or watching, the teacher can be moved to the good and recognize the objective truth contained in the concrete reality. Celebrating liturgies for teachers beautifully makes it more likely that they will be attracted to God and the truth He reveals. Furthermore, formators can point out the truth communicated by the beauty and what the beauty manifests about God.

Incorporating goodness in faith formation likewise attracts teachers and points them to the all-good God. The goodness of human community in fellowship and service can move teachers to love and charity. Not to be overlooked is teachers' prayerful reading of Scripture that can stir affective love of God through the Holy Spirit who inspired it. Teaching *lectio divina*, Liturgy of the Hours, or Ignatian meditation, as well as allowing time for teachers to practice these gives them encounters with God that incline their hearts and strengthen their wills toward the good. Prayerfully reading Scripture, then, sets the stage for teachers' moral formation to engage the discipline of the body and the reasoning mind (*ratio*) to discover and live out goodness. This approach relies on God's presence and grace to assist the teacher to do good, overcoming the effects of the Fall on her.

Including truth in teachers' faith formation can include approaches such as contemplative practice and the truth teachers might recognize through that process. However, formators cannot disregard communicating the objective truths taught by and through the Church. First among these objective truths is the *kerygma*, the principal proclamation of the Gospel which announces God's loving desire to save His people, which firmly orients all faith formation toward teachers' final end of union with God. Teachers' *ratio* can be fruitfully engaged in reasoned explanations of doctrine once the foundation has been established by addressing the *intellectus*. The objective truths of the Faith can be more readily understood by the *ratio* when the whole integrated person of the teacher has been engaged through the *intellectus*, and reasonably explaining these truths prevents errors that arise from using contemplative practice alone as a faith formation tactic.

## 8. Conclusions

Human pedagogical efforts need to take account of the nature of the person who is the subject of these endeavors. The most essential aspect of any such formation for work in Catholic schools can perhaps be summed up best in the idea of *perichoresis*—an acknowledgement of the interplay of every aspect of the person in pursuit of the truth, beauty and goodness of God.

In applying these transcendentals, it is possible to make use of the concrete and material means through which God Himself can form teachers. An analysis of contemplative practice shows that it is insufficient for providing all that a teacher needs to fulfill the mission of the Catholic school. By contrast, an adequate anthropology empowers teachers to fulfill the mission, because it presumes a vision of human learning that accommodates the fullness of human nature as it has been created by God: not just the teacher's bodily nature, not just her *ratio*, but her entire human nature as a spiritual, relational, integrated person who learns and engages in relationship with God through all her powers. Contemplative practice, on the other hand, relies upon key anthropological and educational principles asserted by John Dewey, whose vision heavily influences teacher professional training in the United States. Dewey's ideas, developed from assumptions made by Descartes, Kant, and others, target the *ratio*, which wearies teachers and hardens their hearts toward God. These views are not in keeping with a Catholic understanding of reality and would constitute an inherent contradiction if used in a Catholic context.

Instead, pedagogically creating conditions that facilitate a teacher's encounter with God allows Him to form her whole person, in all her integrated powers, so that she receives objective truth and makes a willing commitment to God. The mission of the Catholic school to educate the whole person for union with God requires that the teacher's whole person—including her body, emotions, will, and intellect—needs to be formed to empower her to fulfill the mission. More importantly, however, the teacher will be formed for her own

final end of union with God. By forming the teacher in her whole person, through fostering analogical encounters with God through the transcendentals and providing sacramental encounters with Him in the liturgy, Catholic schools can fulfill their mission to educate not only students, but teachers, for union with God.

**Author Contributions:** Writing—original draft preparation, A.E.R.; writing—review and editing, G.O. All authors have read and agreed to the published version of the manuscript.

**Funding:** This research received no external funding.

**Institutional Review Board Statement:** Not applicable.

**Informed Consent Statement:** Not applicable.

**Data Availability Statement:** Not applicable.

**Acknowledgments:** The authors would like to thank Christine Forlin of The University of Notre Dame Australia for her editorial assistance and helpful comments.

**Conflicts of Interest:** The authors declare no conflict of interest.

## Notes

1     This paper will use the pronoun "she" in any reference to the human person to more easily distinguish the human person from God.

2     In 2020, the USCCB noted that a number of liturgical hymns manifested these same ten deficiencies, indicating that teachers and students who go to Mass are formed in the faith by music that does not communicate the fullness of the Catholic faith. United States Conference of Catholic Bishops Committee on Doctrine. 2020. Catholic Hymnody at the Service of the Church: An Aid for Evaluating Hymn Lyrics. Available online: https://www.usccb.org/resources/catholic-hymnody-service-church-aid-evaluating-hymn-lyrics (accessed on 29 September 2022).

3     This paper will understand "contemplative practice" as Schuttloffel describes it. "Contemplative practice" in this work does not refer to meditation or contemplation rooted in Eastern mysticism. "Mindfulness" can be understood to encompass more than Schuttloffel appears to imply in her description of "contemplative practice," and so "mindfulness" will not be understood in this paper to be included by Schuttloffel's terminology.

4     In *The Religious Dimension of Education in a Catholic School*, the Congregation for Catholic Education observed that "any genuine educational philosophy has to be based on the nature of the human person, and therefore must take into account all of the physical and spiritual powers of each individual" (Congregation for Catholic Education 1988, #63).

5     Frank Sheed explains "spirit" in humans and in God in the first two chapters of *Theology for Beginners.* In humans, "spirit" is "the element in us by which we know and love" (Sheed [1958] 1981, p. 9) thereby making the "spiritual powers" the powers of intellect and will. God, he states, is infinite spirit (Sheed [1958] 1981, p. 17) and thereby all-knowing and all-loving.

6     All human language about our encounters with God must be referred to as analogical. In the words of Aquinas, "we can know God only from creatures . . . .Thus, whatever is said of God and creatures is said in relation of the creature to God" (Aquinas 1922, I, 13, 5). Following Aquinas, the Catholic Church teaches that human knowledge of God is necessarily limited. This real but inadequate knowledge is referred to as "analogy" (Catechism of the Catholic Church 2009, #40–43). The classic Catholic understanding of the role of analogy comes from Canon 2 of the Fourth Lateran Council, 1215 AD: "Between the Creator and the creature there cannot be a likeness so great that the unlikeness is not greater."

7     Joseph Ratzinger has alluded to this understanding in multiple works, including but not limited to his 2002 message "The Feeling of Things, the Contemplation of Beauty."

8     The authors are indebted to the thesis (A Thomistic Reply to John Dewey's Approach to Learning. Master's Degree of Thomistic Studies, Dominican House of Studies, Washington, DC, USA) of Sr. Mary Thomas Huffman (2019), OP for an understanding of Dewey's ideas.

9     Dewey expressed his belief that humans had "but one sure road of access to truth—the road of patient, cooperative inquiry operating by means of observation, experient, record and controlled reflection" (Dewey 1934, p. 32). This conviction eliminated all possibility of knowing the spiritual.

10     While contemplative practice has been the primary target of this critique, many faith formation efforts that seek to communicate objective truth and Catholic Church teaching likewise fall into the error of hyper-emphasizing the *ratio*. In his book *Poetic Knowledge*, James Taylor observes that after the Council of Trent, faith formators began presenting the Faith as a "demonstration that requires proof" (Taylor 1998, p. 108). The Catholic Church has acknowledged teachers' need for reasoned explanations of Church teaching. But focus on the *ratio* alone fails to address the integrated powers of the human person, and the need to account for all human powers in cultivating teachers' relationship with God. The Church calls for teachers to have "a mastery of the

knowledge of the truths of the faith," as well as "mastery of the . . . principles of spiritual life that require constant improvement" (Congregation for Catholic Education 2007, #26).

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
