# Peer review of "The Integral Formation of Catholic School Teachers"

_religions, doi:10.3390/rel13121230_

Round 1

Reviewer 1 Report

I think that the introduction of intellectus into the educational debate is very important.

The two main areas in which this article can be improved are in the definition of terms and more referencing. More terms need to be defined since I think that the author assumes knowledge on the part of potential readers. Not everyone is a Thomist or scholastic. They may not even have a good knowledge of Catholic theology. The terms I would like to see defined, or defined more fully, are faith (is it faith in God, what God reveals, or doctrine?); perichoresis, sensitive powers, sensitive appetite (when the term is first used), transcendentals, spiritual powers, analogical and analogical encounters, and the Fall.

References are needed to Pieper and Ratzinger (198) (also, it is Joseph, not Josef); Montessori, Cavaletti and Gobbi (perhaps a reference to the Catechesis of the Good Shepherd); and to Aquinas and Balthasar (386).

Finally, a few minor points. In the list "body, spirit, emotions, will, intellect", I would leave out "spirit", since the will and intellect will later be called "spiritual powers".

In what way to the Trinitarian persons "seek" communion with each other? (146) Perhaps this could be rephrased.

I think an example of how the intellectus actually "works" would help the reader understand it (308-311). 

Author Response

Thank you for your feedback and the opportunity to make revisions.

The terms I would like to see defined, or defined more fully, are faith (is it faith in God, what God reveals, or doctrine? See lines 64-67 and citation added to Bibliography); perichoresis added 202-204 and endnote vi with bibliographic citation, sensitive powers added lines 195-196, sensitive appetite (when the term is first used) added lines 197-198, transcendentals added lines 400-401 with citation and Bibliography, spiritual powers added lines 154 and endnote v plus citation in Bibliography, analogical and analogical encounters added 206-207 and endnote vi with bibliographic citation, and the Fall added 123-127 and citation added to Bibliography.

References are needed to Pieper and Ratzinger (198) - added in line 215 and endnote v (incl. one example of Ratzinger) (also, it is Joseph, not Josef -corrected); Montessori, Cavaletti and Gobbi (perhaps a reference to the Catechesis of the Good Shepherd) added in 407 and 410 and Bibliography; and to Aquinas and Balthasar (386 now 434) added in 437.

Finally, a few minor points. In the list "body, spirit, emotions, will, intellect", I would leave out "spirit", since the will and intellect will later be called "spiritual powers". Deleted "spirit" from the relevant lists.

In what way to the Trinitarian persons "seek" communion with each other? (146) Perhaps this could be rephrased. "Seek" changed to "in."

I think an example of how the intellectus actually "works" would help the reader understand it (308-311). An illustration has been added in lines 238-257.

Reviewer 2 Report

This paper makes a valuable contribution to debates about the nature of the formation of teachers for Catholic schools in the US and offers a theoretical framework for the inclusion of practices beyond just the cognitive domain alone. The author is realistic and recognises the need for faith formation of Catholic teachers in the context of the growth of secularisation in the US and acknowledges the limitations of Schuttloffel’s work, an advocate for contemplative practice, which the author says is predicated on teachers having been already rooted and formed in objective Church teaching and practice. The pervasive influence of John Dewey on the educational system in the US is bemoaned and the author provides a critique of Dewey's anthropological assumptions, which deny the importance of the spiritual dimension of the person. Section 6 of the paper is particularly well articulated. In this section, the author outlines an integrated approach drawing on the intellectus in the first instance which leads to the ratio domain being stimulated. The paper them proposes a way forward by applying the transcendentals to teacher faith formation and cites the work of Maria Montessori and her advocates as exemplars in this regard.

One aspect of the article which merits a second appraisal is the treatment of contemplative practice.

1.    Firstly there is the lack of a clear definition of contemplative practice.  Within broader educational contexts, the area of contemplative pedagogies and deep reflection has grown in scholarship and practice and perhaps some key theorists in this area could be cited (e.g. Orwen Ergas).

2.    Secondly, Schuttloffel’s work, which is cited as the main source, dates back to 2013 and Fussell’s article, although more recent, doesn’t properly address contemplative practice apart from naming it as an example of a practice used by some of the interviewee teachers in his study. The main point emerging in this paper from the reference to Schuttloffel is the recognition of the need for ‘foundational formation’ and the over-optimism of Schuttloffel with regard to the capacity teachers in formation to engage with contemplative practices.

3.    It’s not clear how Schuttloffel’s practice of contemplation differs from that suggested in terms of introducing students to an experience of the transcendentals.

Overall, this paper is very well presented. It is thought provoking and timely as the nature of Catholic teacher formation programmes are being reviewed in light of societal changes and expectations of stakeholders.  

Typographical Errors / Suggestions for rewording

Line 12- Dewey’s anthropological principles are referred to as ‘faulty’. Could this be amended to ‘inadequate’ which seems more measured and which is inferred in the article also.

Line 26 – typo. Repetition of the work ‘this’.

Line 34 – ‘Evangelisation’ is spelt here with an ‘s’ and with a ‘z’ elsewhere. Consistency required.

Line 104 – ellipsis …. Perhaps close inverted commas at this point and paraphrase the next sentence as its current expression doesn’t flow well.

Line 106 – place a comma after ‘is’ at the end of the line.

Author Response

Thank you for your feedback and the opportunity to respond and revise.

One aspect of the article which merits a second appraisal is the treatment of contemplative practice.

  1. Firstly there is the lack of a clear definition of contemplative practice.  A definition is added from a 2016 article by Schuttloffel, which has also been added to the Bibliography. Within broader educational contexts, the area of contemplative pedagogies and deep reflection has grown in scholarship and practice and perhaps some key theorists in this area could be cited (e.g. Orwen Ergas). Endnote iii has been added to clarify that Schuttloffel's contemplative practice is distinct from other contemplative pedagogies which are rooted in Eastern mysticism rather than Catholicism.  
  2. Secondly, Schuttloffel’s work, which is cited as the main source, dates back to 2013 and Fussell’s article, although more recent, doesn’t properly address contemplative practice apart from naming it as an example of a practice used by some of the interviewee teachers in his study. The purpose of citing Fussell is to show that Schuttloffel's contemplative practice continues to be used by Catholic school teachers today - see lines 96-103 which have been revised to better show this. Also, the National Catholic Education Association website (www.ncea.org) continues to sell Schuttloffel's works with no other faculty faith formation materials evidently available...her work continues to be disseminated through NCEA. The main point emerging in this paper from the reference to Schuttloffel is the recognition of the need for ‘foundational formation’ and the over-optimism of Schuttloffel with regard to the capacity teachers in formation to engage with contemplative practices. Perhaps the main point could be rephrased this way: "...the over-optimism of Schuttloffel with regard to teachers' ability to evaluate the results of their contemplative practice in light of Catholic Church teaching." Hopefully the revised paragraph more clearly expresses this intended point.
  3. It’s not clear how Schuttloffel’s practice of contemplation differs from that suggested in terms of introducing students to an experience of the transcendentals. Lines 407-417 added to clarify this.

Typographical Errors / Suggestions for rewording

Line 12- Dewey’s anthropological principles are referred to as ‘faulty’. Could this be amended to ‘inadequate’ which seems more measured and which is inferred in the article also. Changed to "inadequate."

Line 26 – typo. Repetition of the work ‘this’. Deleted repeated word.

Line 34 – ‘Evangelisation’ is spelt here with an ‘s’ and with a ‘z’ elsewhere. Consistency required. Changed to "Evangelization."

Line 104 – ellipsis …. Perhaps close inverted commas at this point and paraphrase the next sentence as its current expression doesn’t flow well. Broke into 2 separate sentences to make it flow better while still retaining Schuttloffel's wording.

Line 106 – place a comma after ‘is’ at the end of the line. Comma added.